# Soft-Unification in Deep Probabilistic Logic

**Jaron Maene**
KU Leuven
jaron.maene@kuleuven.be

**Luc De Raedt**
KU Leuven & Örebro University
luc.deraedt@kuleuven.be

## Abstract

A fundamental challenge in neuro-symbolic AI is to devise primitives that fuse the logical and neural concepts. The Neural Theorem Prover has proposed the notion of soft-unification to turn the symbolic comparison between terms (i.e. unification) into a comparison in embedding space. It has been shown that soft-unification is a powerful mechanism that can be used to learn logic rules in an end-to-end differentiable manner. We study soft-unification from a conceptual point and outline several desirable properties of this operation. These include non-redundancy in the proof, well-defined proof scores, and non-sparse gradients. Unfortunately, these properties are not satisfied by previous systems such as the Neural Theorem Prover. Therefore, we introduce a more principled framework called DeepSoftLog based on probabilistic rather than fuzzy semantics. Our experiments demonstrate that DeepSoftLog can outperform the state-of-the-art on neuro-symbolic benchmarks, highlighting the benefits of these properties.

## 1 Introduction

Deep learning has become the most prominent paradigm in machine learning but still suffers from limitations such as poor interpretability, reasoning skills, and difficulties in incorporating knowledge. Neuro-symbolic AI (NeSy) is an emerging field that seeks to address these limitations by combining the strengths of neural networks and symbolic systems [14, 16].

A key design question in NeSy is the used *representation* [14, 16]. Neural networks operate on tensors while symbolic reasoning operates on symbolic structures. Previous systems have broadly used two ways to deal with this dilemma. The first is keeping the neural and symbolic representations completely separate (e.g. DeepProbLog [22]). The second is relaxing the symbolic part, making it continuous by mapping the symbols onto vector spaces (e.g. Logic Tensor Networks [4]). This allows – in principle – for a tighter integration, where both representations are retained and fused.

The Neural Theorem Prover (NTP) [29] is an early system that pioneered such an integration, by linking the logical and neural representations with a mechanism called soft-unification. For instance, in regular logic, matching `grandpa(jef, john)` with `grandfather(jef, john)` or `event(earthquake)` and `event(landslide)` fails. However, by mapping `grandfather` and `grandpa`, resp. `earthquake` and `landslide` to vectors, they can be compared in an embedding space. The NTP is an end-to-end differentiable prover that learns these embeddings from data.

Inspired by the success of the NTP, we analyze its key concept – *soft-unification* – from a theoretical perspective. We identify several natural and desirable properties of soft-unification, which are not always satisfied by previous works. This motivates us to propose an alternative more principled definition of soft-unification based on distances. Next, we introduce DeepSoftLog, which makes a principled integration of embeddings in probabilistic logic programming [10].

As an example, consider the DeepSoftLog program in listing 1. By associating the constants with learnable embeddings (denoted with the $\sim$ prefix), we turn a regular finite state machine

37th Conference on Neural Information Processing Systems (NeurIPS 2023).

implementation into a differentiable model. So if the program would be trained on e.g. the $(01)^*$ language, it could learn to set $\sim$`prev_state1` = $\sim$`state2` and $\sim$`prev_state2` = $\sim$`state1`.

In summary, this paper makes three contributions: (1) a theory for using learnable embeddings in logic through soft-unification, (2) DeepSoftLog, a neuro-symbolic system that includes embeddings inside an end-to-end probabilistic logic prover, and (3) a demonstration of the practical advantages of DeepSoftLog over state-of-the-art NeSy systems on several benchmarks.

```
accepts(X) :- run(~start_state, X).
run(~end_state, []).  % base case + 2 state transitions
run(~state1, [~symbol1|T]) :- run(~prev_state1, T).
run(~state2, [~symbol2|T]) :- run(~prev_state2, T).
query(accepts([~0, ~1])).
```

Listing 1: Example of a differentiable finite state machine implementation in DeepSoftLog.

## 2 Background

### 2.1 Logic programming

We summarise the basic logic programming concepts. An in-depth discussion can be found in [13].

**Syntax** A term `t` is either a constant `c`, a variable `X`, or a structured term `f(t`$_1$`, ..., t`$_k$`)`, where `f` is a functor and `t`$_i$ are terms. Atoms are expressions of the form `q(t`$_1$`, ..., t`$_k$`)`, where `q` is a predicate of arity $k$ and `t`$_i$ are terms. A rule is an expression `h :− b`$_1$`, ..., b`$_k$ where `h` is an atom and `b`$_i$ are atoms or negated atoms. The meaning of such a rule is that `h` holds whenever all the `b`$_i$'s hold. The atom `h` is the head of the rule, while the conjunction of `b`$_i$'s is the body. Facts are rules with an empty body.

A logic program is a set of rules. An expression is *ground* if it does not contain variables. To ground an expression `e` we can apply a substitution $\theta = \{V_1 \rightarrow t_1, ..., V_k \rightarrow t_k\}$, creating an instantiated expression `e`$\theta$ where every occurrence of the variable `V`$_i$ is replaced with `t`$_i$. An important operation is *unification*. Two expressions `t`$_1$ and `t`$_2$ unify when there exists a substitution $\theta$ such that $t_1\theta = t_2\theta$. For example, `f(g(a), X)` and `f(g(Y), Z)` have a unifier $\{X \rightarrow Z, Y \rightarrow a\}$.

**Semantics** The set of all possible ground terms in a program is called the *Herbrand universe* $\mathcal{U}$. A *possible world* (or Herbrand interpretation) is a subset of the possible ground atoms. A ground atom $a$ is true in a possible world $w$ when $a \in w$. A possible world $w$ is a model for a program $T$, written $w \models T$, when every grounding of every rule in the program is true in the possible world.

**Inference** Proving relies on the repeated application of rules to a goal, using *SLD resolution*. Given a rule `h :− b`$_1$`, ... b`$_n$ and a goal `g`$_1$`, ... g`$_m$, such that `h` and `g`$_1$ unify with substitution $\theta$, SLD resolution derives the new goal `(b`$_1$`, ... b`$_n$`, g`$_2$`, ... g`$_m$`)`$\theta$. A proof is a sequence of resolution steps that results in an empty goal.

### 2.2 Probabilistic and Fuzzy Logic

ProbLog [10] extends logic programming with probabilistic facts `p :: f`, where a ground fact `f` is annotated with a probability `p`. As an example, consider the well-known alarm Bayesian network:

```
0.1 :: event(landslide).  0.2 :: event(earthquake).
0.5 :: hears_alarm(mary).  0.4 :: hears_alarm(john).
alarm :− event(landslide).  alarm :− event(earthquake).
calls(X) :− alarm, hears_alarm(X).
```

**Semantics** Each ground probabilistic fact represents an independent Boolean random variable. Hence, the probability of a possible world $w$ is a product over the choice of probabilistic facts in $w$. The probability (or *success score*) of an atom `a` in a program $T$ is the sum over all the models of $T$ where `a` is true.

$$P(w) = \prod_{f \in w} p_f \prod_{f \notin w} (1 - p_f) \qquad\qquad P(\mathtt{a}) = \sum_{w : \mathtt{a} \in w \land w \models T} P(w)$$

In the example, a choice of probabilistic facts $F = \{\texttt{event(landslide)}, \texttt{hears\_alarm(mary)}\}$, has the corresponding model $w = F \cup \{\texttt{alarm}, \texttt{calls(mary)}\}$ with probability $P(w) = 0.1 \times 0.5 \times (1 - 0.2) \times (1 - 0.4) = 0.024$.

**Inference** There are multiple ways to perform inference in ProbLog [12]. We rely on a proving approach [24]. Computing the success score $P(q)$ of a query $q$ consists of the following steps. (1) Generate the set of all proofs $B(q)$ using SLD resolution. (2) Disjoin the proofs into a formula $\bigvee_{\pi \in B(q)} \pi$. (3) Transform this formula using knowledge compilation [7], to ensure it can be evaluated correctly and efficiently. (4) Convert the compiled formula into an arithmetic circuit by replacing the OR and AND operations with SUM and PRODUCT operators from a semiring [20].

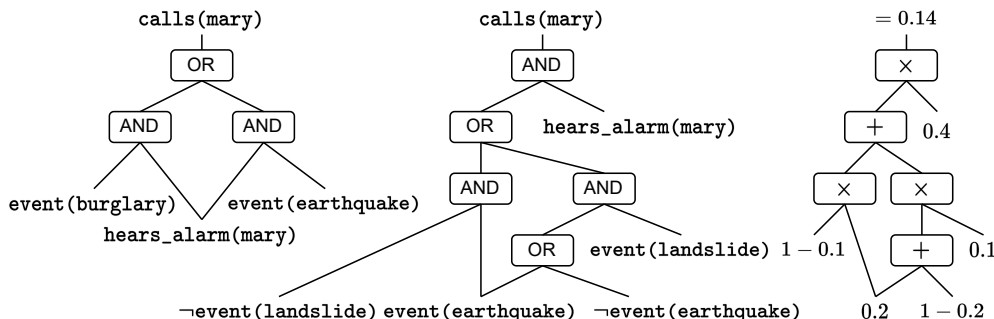

Figure 1: The circuits used during inference of $\texttt{calls(mary)}$ in the alarm example. (left) AND/OR formula represented as a tree. (middle) Compiled formula. (right) Arithmetic circuit.

We demonstrate our inference approach on the query $\texttt{calls(mary)}$. The proofs: $\texttt{event(earthquake)} \wedge \texttt{hears\_alarm(mary)}$ and $\texttt{event(landslide)} \wedge \texttt{hears\_alarm(mary)}$ result in the formula on the left of figure 1 when disjoined. When working with probabilities, we cannot simply evaluate this formula. Indeed, the proofs are not independent as they both contain $\texttt{hears\_alarm(mary)}$, so $P(A \vee B) \neq P(A) + P(B)$. Hence, we transform the formula into a circuit using knowledge compilation [7] (middle of figure 1). The resulting circuit supports many types of inference, by evaluating with the appropriate semiring [19]. For probabilistic inference, we can simply use the standard $+$ and $\times$ operators, as shown on the right of figure 1.

**Neural Theorem Prover** We can also sketch the Neural Theorem Prover (NTP) [29] in this framework. The NTP is based on the Datalog subset of Prolog. So it does not include structured terms (e.g. lists) or negation. At the same time, rather than using the probabilistic semiring, the NTP uses the fuzzy semiring. Essentially, this means the fuzzy $\max$ and $\min$ operations are used instead of the probabilistic $+$ and $\times$. This is also known as the Gödel t-norm or minimum t-norm. In contrast with probabilistic inference, disjunction in the fuzzy semiring is idempotent. This implies we can skip the expensive knowledge compilation step and evaluate the formula directly during proving [19].

**Approximation** As probabilistic inference is #P-hard, it is possible to approximate inference by only using the $k$-best proofs. The best proofs are those with the highest score. During proving, these can be found with $A^*$ search on the proof tree [24]. For instance, the 1-best proof for the example is $\texttt{event(earthquake)} \wedge \texttt{hears\_alarm(mary)}$, with a score of $0.08$. Note that for the NTP, the 1-best proof captures the full success score due to the $\max$ aggregation of proofs. The NTP and Greedy NTP [26] introduced additional approximation methods to prune proofs during proving, such as imposing a maximum depth and maximum branching factor.

### 2.3 Soft Unification

While in standard logic programming two constants $\texttt{landslide}$ and $\texttt{earthquake}$ do not unify (i.e. do not match), soft-unification returns a score based on how similar the symbols are in embedding space. We use the $\simeq$ predicate to denote that two symbols are soft-unified (e.g. $\texttt{earthquake} \simeq \texttt{landslide}$). The score of a soft-unification is determined by a soft-unification function $s : \mathcal{S} \times \mathcal{S} \rightarrow [0, 1]$, so $P(t_1 \simeq t_2) = s(t_1, t_2)$.

Algorithm 1 states the simplified soft-unification algorithm for atoms and terms without functors. The algorithm returns the set of variable substitutions, that make two atoms equal while disregarding the

equality of the symbols. For example, the two atoms $p(a, b, X, Y)$ and $p(c, d, Z, d)$ do not unify, but soft-unify with the substitution $\{X \to Z, Y \to d\}$. In practice, the algorithm usually also extracts the conjunction of the used soft-unifications (in this example $(a \simeq c) \wedge (b \simeq d)$). During inference, these can be conjoined with the proof just like probabilistic facts.

---

**Algorithm 1** Soft-unification

---

**function** SOFTUNIFY$(t_1, t_2)$
    **if** $t_1$ is a variable **then return** $\{t_1 \to t_2\}$
    **else if** $t_2$ is a variable **then return** $\{t_2 \to t_1\}$
    **else if** $t_1$ and $t_2$ are constants **then return** $\varnothing$
    **else if** $t_1$ and $t_2$ are atoms of the same arity $k$ **then**
        **return** $\bigcup_{i \in \{1..k\}}$ SOFTUNIFY(argument $i$ of $t_1$, argument $i$ of $t_2$)
    **else return** Failure
    **end if**
**end function**

---

To illustrate soft-unification during proving, suppose that $s(\texttt{landslide}, \texttt{earthquake}) = 0.5$. We can now delete the fact `event(landslide)` from our example program and still derive the same probability for the query `calls(mary)`. The first proof for `event(landslide)` (namely `event(earthquake)` $\wedge$ `hears_alarm(mary)`) is unaffected by the deletion of `event(landslide)`. But in the second proof, we need to derive `event(landslide)` by applying `event(earthquake)`, resulting in the proof `event(earthquake)` $\wedge$ `(landslide` $\simeq$ `earthquake)` $\wedge$ `hears_alarm(mary)`.

## 3 Soft-unification properties

In principle, any function $s$ can be used for soft-unification. However, we introduce some properties that make the soft-unification semantically meaningful and efficiently trainable.

**Definition 1.** *The success score $P(q)$ is well-defined iff for every soft-unification function $s$, query $q$, and pair of logically equivalent programs $T_1$ and $T_2$ (i.e. $\forall a : T_1 \models a \Leftrightarrow T_2 \models a$), the success score $P(q)$ is the same for $T_1$ and $T_2$.*

**Definition 2.** *The soft-unification function $s$ is non-redundant, when given a proof $\pi$, every other entailed proof of the form $\pi' = (\pi \backslash \{t_i \simeq t_j\}) \cup \{t_i \simeq t_k, t_k \simeq t_j\}$ has at most the same score: $P(\pi') \leq P(\pi)$.*

We motivate definition 2 by returning to our running example. Recall that we made a proof where `event(landslide)` was obtained using the soft-unification: `(landslide` $\simeq$ `earthquake)`. Suppose now that the program contains some extra rules: `alarm :− warning(landslide)` and `warning(avalanche) :− event(avalanche)`. These rules could be interjected in the proof so that we use: `(landslide` $\simeq$ `avalanche)` $\wedge$ `(avalanche` $\simeq$ `earthquake)`. Definition 2 says that introducing these intermediary soft-unifications in the proof should not increase the proof score.

**Definition 3.** *The soft-unification function $s$ is connected, when for every $x$ and $z$ where $x \neq z$, it is possible to have a $y$ that is inbetween $x$ and $z$ (i.e. $s(x, y) > s(x, z)$ and $s(y, z) > s(x, z)$).*

Definition 3 is a statement on the expressivity of soft-unification. For example, it should be possible to model that white and grey are similar (e.g. $s(\texttt{white}, \texttt{grey}) = 0.5$), and grey and black are similar (e.g. $s(\texttt{grey}, \texttt{black}) = 0.5$), but that white and black are dissimilar (e.g. $s(\texttt{white}, \texttt{black}) = 0.25$).

**Definition 4.** *We call the proof score $P(q)$ effectively optimizable when the soft-unification $s$ is differentiable, and it is possible to have a gradient with respect to every proof $\pi$ in $B(q)$.*

This last definition essentially asks that the soft-unification function should be trainable using gradient descent. It is important that every proof receives gradients, as updating a single proof at a time leads to local minima due to a lack of exploration [9, 24]. In short, if a wrong proof starts of with a higher score due to an unfortunate initialization, it should not get pushed up greedily disregarding the other proofs. Otherwise, the correct proof might never be discovered during training. Note that $k$-best approximate inference does not evaluate all possible proofs and hence cannot satisfy definition 4.

| Properties | Def. 1 | Def. 2 | Def. 3 | Def. 4 |
|---|---|---|---|---|
| Bousi∼Prolog [17] | Yes | Depends (*) | Depends (*) | No |
| WHIRL [6] | No | No | Yes | No |
| NTP [29] | No | No | Yes | No |
| LRNN [31] | No | No | Yes | Yes |
| DeepSoftLog (Ours) | Yes | Yes | Yes | Yes |

Table 1: Comparison of the soft-unification properties of different (neuro-)symbolic systems. (*): Depends on the choice of transitivity and t-norm.

### 3.1 Neural Theorem Prover

The current (neuro)-symbolic systems unfortunately do not always respect the abovementioned properties. We focus mostly on the Neural Theorem Prover, though table 1 gives an overview of several related (neuro-)symbolic systems, which we later discuss in the related work. We start by introducing some necessary concepts, before giving the proof that the NTP is not always well-defined.

An atom or term is *linear* when no variable occurs multiple times. A rule is linear when its head is linear. A soft-unification function $s$ is $\wedge$-*transitive* when $s(x,z) \geq s(x,y) \wedge s(y,z)$, and a $\wedge$-*similarity* when it is reflexive, symmetric, and $\wedge$-transitive. The specific meaning of the conjunction operation $\wedge$ depends on the semantics. So $\wedge$-transitivity should be interpreted as $\min$-transitivity for the NTP and $\times$-transitivity for ProbLog. Omitted proofs can be found in appendix A.

**Theorem 1.** *The Neural Theorem Prover is not well defined on programs with linear rules.*

*Proof.* We give a proof by example. Consider the following two programs:
eq(X, X) :− p(X). p(a). p(b). p(c). and eq(a, a). eq(b, b). eq(c, c). p(a). p(b). p(c).
These programs are logically equivalent but can have a different proof score for the query $q = $ eq(a, b). In $T_1$, we obtain $P(q) = s(\mathtt{a}, \mathtt{b})$ in the straightforward manner. But in the second proof, we obtain $P(q) = \max(s(\mathtt{a}, \mathtt{b}), \min(s(\mathtt{a}, \mathtt{c}), s(\mathtt{b}, \mathtt{c})))$, because we can also unify with eq(c, c). So with the right choice of $s$, the proof score $P(q)$ is higher in $T_2$ than in $T_1$. □

This proof works in general for programs that contain linear rules. It might be tempting to suggest a fix by making $s$ $\wedge$-transitive, but this does not resolve the problem when negation is considered. Transitivity is still desirable for another reason however, as it relates to definition 2.

**Theorem 2.** *Definition 2 holds when the soft-unification function $s$ is $\wedge$-transitive.*

We note that the NTP implements the soft-unification function $s$ using radial basis functions [5] (specifically the laplacian kernel $e^{-\|x-y\|_2}$), which are not $\min$-transitive. The situation in the NTP is further complicated as the choice of semantics links definition 2 and 3.

**Theorem 3.** *Under the fuzzy semiring, definition 3 cannot hold if $s$ is $\wedge$-transitive.*

In other words, by choosing the minimum t-norm, definitions 2 and 3 are mutually exclusive. A further disadvantage is that the minimum t-norm also does not satisfy definition 4. By using the minimum to conjoin facts, the success score of a proof is essentially determined by the weakest soft-unification. This is a significant drawback as it heavily reduces training efficiency and leads to local minima [9].

### 3.2 ProbLog

Motivated by the previous discussion, we will rely on probabilistic instead of fuzzy semantics. This allows us to satisfy all the stated properties.

**Theorem 4.** *If the soft-unification function $s$ is a $\times$-similarity with probabilistic semantics, we can satisfy all stated properties (from Def. 1, Def. 2, Def. 3 and Def. 4).*

**Theorem 5.** *A soft-unification function $s$ is a $\times$-similarity, iff $s(x,y) = e^{-d(\mathcal{E}(x),\mathcal{E}(y))}$, where $d$ is a distance function on an embedding space and $\mathcal{E}$ maps symbols to this space.*

The last theorem shows that the soft-unification corresponds to a distance on a embedding space under the modest assumptions of reflexivity, symmetry, and transitivity. This brings us to the final question of what concrete embedding space to choose. We do this based on two considerations.

This first is the gradient norm of the soft-unification. It is undesirable to have large regions of near-zero gradients for many successive soft-unification steps (also known as the *vanishing gradient* problem). Due to the exponential function, this can only be achieved by taking a bounded domain. We implement this by choosing a hypersphere as the embedding space, i.e. the set of normalized vectors $\{x \in \mathbb{R}^n : \|x\|_2 = 1\}$.

A natural distance to use on hyperspheres is the angle. A drawback of using the angle is that two symbols on the opposites of the hypersphere are maximally dissimilar. This is undesirable as we cannot have many symbols that are all dissimilar to each other. To solve this we use $s(x, y) = \exp(-\lambda \arccos|\mathcal{E}(x) \cdot \mathcal{E}(y)|)$ as the soft-unification. The $\lambda$ is a scaling factor, which can be set based on the desired minimal soft-unification score. The absolute value assures that maximum dissimilarity is achieved at 90 degrees instead of 180. So as random high-dimensional vectors are almost surely orthogonal, soft-unification scores at initialization will be zero in expectation.

# 4 DeepSoftLog

The previous section discussed the combination of soft-unification with probabilistic semantics. We will now formally implement it as an extension of ProbLog, called DeepSoftLog.

**Syntax** DeepSoftLog is a superset of the probabilistic logic programming language ProbLog. This means that every valid ProbLog program is also a valid DeepSoftLog program. Syntactically, DeepSoftLog only adds a single syntactic element: embedded terms.

**Definition 5.** *An embedded term is a term* $\sim$t*, where* $\sim$/1 *is the embedding functor and* t *is a term.*

The embedding functor turns a term t into an embedded term $\sim$t. As the embedding functor is unary, we do not write brackets. The embedding functor is idempotent, and applying it inside an embedded term has no effect. For example, $\sim\sim$f($\sim$X) is equivalent to $\sim$f(X). Embedded terms can be composed like any other term, e.g. f($\sim$a, g(X, $\sim$h(b))). The embeddings are computed using a dictionary $\mathcal{D}$ which stores a vector $\mathcal{D}(c) \in \mathbb{R}^n$ for all constants c and a neural network $\mathbb{R}^{k \times n} \to \mathbb{R}^n$ for every functor f of arity $k$. The embedding of a ground term $\mathcal{E} : \mathcal{U} \to \mathbb{R}^n$ is recursively defined as $\mathcal{E}(c) = \mathcal{D}(c)$ for a constant c and $\mathcal{E}(f(t_1, ...t_k)) = \mathcal{D}(f)(\mathcal{E}(t_1), ...\mathcal{E}(t_k))$ for a structured term.

We note that we deviate from the NTP in several important regards concerning what we embed. (1) The embedding of terms is optional. This means the programmer is free to choose, where it makes sense to rely on embeddings, and what to keep symbolic. This can help keep inference tractable, as letting everything soft-unify leads to huge branching factors. (2) DeepSoftLog does not embed predicate symbols. However, this is not an actual restriction, and the NTP behavior can easily be simulated. For example, a(x). b(y). in NTP notation turns into r($\sim$a, $\sim$x). r($\sim$b, $\sim$y). in DeepSoftLog notation. (3) DeepSoftLog can embed structured terms. Embedded functors map embeddings to new embeddings and are a natural place to use neural networks.

**Definition 6.** *A **DeepSoftLog program** $T$ is a tuple $(s, T_{\mathcal{F}}, T_{\mathcal{R}})$. The soft-unification function $s : \mathcal{U} \times \mathcal{U} \to [0, 1]$ maps two embedded ground terms to the probability that they unify. $T_{\mathcal{F}}$ is a set of ground probabilistic facts and $T_{\mathcal{R}}$ a set of linear rules. The atoms in $T_{\mathcal{F}}$ and $T_{\mathcal{R}}$ can contain embedded terms.*

**Semantics** We base the semantics of DeepSoftLog on ProbLog, by creating a source-to-source translation. The basic idea is to eliminate the embedded terms by simulating the soft-unification with a predicate $\simeq$. More concretely, all embedded terms in the head of a rule are replaced by a new unique variable. Next, $\simeq$ atoms are added at the start of the body to provide the soft-unification probabilities. An example of this transformation, p($\sim$x) :- q($\sim$y). becomes p($\sim$V$_1$) :- $\sim$V$_1 \simeq \sim$x, q($\sim$y).

After applying this transformation to all rules in $T_{\mathcal{R}}$ and $T_{\mathcal{F}}$, we obtain a regular ProbLog program. The soft-unification function itself can also be encoded in ProbLog by adding facts for all pairs of ground terms: p :: $\sim$t$_1 \simeq \sim$t$_2$, where p $= s(t_1, t_2)$. We provide some examples of this transformation on full programs in appendix B. We note that the resulting ProbLog program can be infinite, by grounding out the soft-unification predicate $\simeq$ on an infinite set of ground terms. However, the ProbLog semantics are still well-defined on infinite programs, due to the use of Sato's distribution semantics [30].

We note that this program transformation is only for semantics, and we still apply the soft-unification algorithm during inference. In appendix C, we prove that the soft-unification algorithm is equivalent to regular unification on the transformed program.

**Non-linear rules** The semantics of DeepSoftLog only allow for linear rules, but non-linear rules can be supported as syntactic sugar. Similar to Bousi$\sim$Prolog [18], we linearize rules as follows. The variable $X$ that occurs multiple times in the head is replaced by unique variables, and $\simeq$ predicates are added to the body to perform the soft-unification. For example, the linearization of $p(X, X) :- q(X)$. is $p(V_1, V_2) :- V_1 \simeq V_2, q(V_1)$. This construction does entail some extra grounding work on $\simeq$. We need to state the $c \simeq c$ fact for every constant $c$, to make sure $p(X, X)$ in this example can still succeed with non-embedded constants. Similarly, we need rules $f(V_1, ...V_k) \simeq f(W_1, ...W_k) :- V_1 \simeq W_1, ...V_k \simeq W_k$ for every functor $f$ of arity $k$.

**Inference** We use proof-based inference, as described in section 2.2. Exact inference can become intractable for large experiments, so we support approximate inference techniques ($k$-best, maximum search depth, and maximum branching factor), similar to the Greedy NTP [26]. An important constraint we apply is that the soft-unification predicate $\simeq$ needs to be ground upon evaluation, i.e. input mode (++,++) in Prolog. Otherwise, the number of possible unifications explodes. For example in $a \simeq f(X)$, $X$ can unify with an infinite number of ground facts: $a \simeq f(a), a \simeq f(f(a)), a \simeq f(f(f(a))), ...$

During training, we can use the gradient semiring of algebraic ProbLog to make the success score end-to-end differentiable [19]. We train using the AdamW optimizer [21] with cross-entropy loss.

**Structure learning** As in the NTP, we can add templates to the program to do rule learning via parameter learning. For example, $r(\sim t_1, X, Y) :- r(\sim t_1, X, Z), r(\sim t_1, Z, Y)$. is a transitive template that could be used for the facts $r(\sim siblings, alice, bob).r(\sim siblings, bob, charlie)$. by learning that $\sim siblings = \sim t_1$. These templates can also parameterize the rule symbol $t_1$, similar to the conditional theorem prover [27]: $r(\sim T, X, Y) :- r(\sim f_1(T), X, Z), r(\sim f_2(T), Z, Y)$.

## 5 Experiments

We present three sets of experiments. First, we compare DeepSoftLog to the NTP on *structure learning* tasks, which requires learning rules from data. We show that DeepSoftLog significantly improves the performance over the NTP, and perform ablations to show that the properties introduced in section 3 are instrumental to this. The second experiment requires *learning the perception*, while the background knowledge is given. The last experiment is the most challenging setting, as both perception and rules need to be learned *jointly*. For all experiments, we report the mean and standard deviation over 10 seeds. All code is available on `https://github.com/jjcmoon/DeepSoftLog`.

### 5.1 Countries

Countries is a knowledge graph with the `locatedIn` and `neighborOf` relations between the countries and continents on Earth. There are three tasks of increasing difficulty (S1, S2 and S3). For example in S2, we need to derive `locatedIn(france, europe)` using facts such as `locatedIn(belgium, europe)` and `neighborOf(france, belgium)`. For the full experimental setup, we refer to previous work [26, 29]. To keep the comparison fair, we use the same rule templates, embedding dimensions, and training objective as the NTP. The hyperparameters can be found in appendix D.2. Table 2 reports the results and compares them with other state-of-the-art NeSy systems that support structure learning.

In figure 2, we perform ablation on the properties of DeepSoftLog, discussed in section 3. As expected, DeepSoftLog no longer properly converges when switching from probabilistic semantics to fuzzy semantics (violating property 4). Switching the soft-unification function $s$ to a Gaussian kernel (violating property 2) also delays convergence. There is no ablation for property 3 as there is no straightforward way to violate it in DeepSoftLog.

### 5.2 MNIST-addition

MNIST-addition is a popular neuro-symbolic benchmark where, given two numbers as lists of MNIST images, the task is to predict the sum (e.g.  +  = 135). An attractive feature of this task is

| Countries | S1 | S2 | S3 |
|---|---|---|---|
| NTP [29] | $90.93 \pm 15.4$ | $87.40 \pm 11.7$ | $56.68 \pm 17.6$ |
| GNTP [26] | $99.98 \pm 0.05$ | $90.82 \pm 0.88$ | $87.70 \pm 4.79$ |
| DeepSoftLog (Ours) | $\textbf{100.0} \pm 0.00$ | $\textbf{97.67} \pm 0.98$ | $\textbf{97.90} \pm 1.00$ |
| NeuralLP [34] | $\textbf{100.0} \pm 0.0$ | $75.1 \pm 0.3$ | $92.2 \pm 0.2$ |
| CTP [27] | $\textbf{100.0} \pm 0.00$ | $91.81 \pm 1.07$ | $94.78 \pm 0.0$ |
| MINERVA [8] | $\textbf{100.0} \pm 0.00$ | $92.36 \pm 2.41$ | $95.10 \pm 1.2$ |

Table 2: AUC-PR on Countries link prediction [29]. Results are adapted from [27]. The three methods on top use templates, while the bottom ones do not.

| Ablations | VP | AUC-PR |
|---|---|---|
| None | | $\textbf{97.67} \pm 0.98$ |
| Gödel t-norm | 2, 4 | $23.73 \pm 4.08$ |
| Gaussian kernel | 2 | $28.33 \pm 3.64$ |

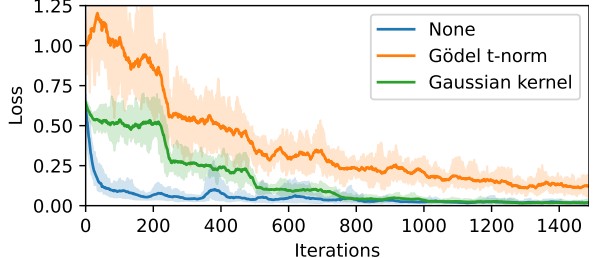

Figure 2: (Left): Ablations of DeepSoftLog on countries S2, with the corresponding properties that are violated. VP stands for violated properties. (Right): Loss curve during training of the ablations. The shaded area represents the min/max over 10 seeds, while the line is the mean.

that it allows to increase the reasoning difficulty, by increasing the number of digits. We use the same experimental setup and neural network as previous works, to which we refer for more details [22].

Existing NeSy solutions to the MNIST-addition problem typically represent all the possible sums explicitly, which means inference becomes exponential in the number of digits. DeepSoftLog allows an alternative encoding of the problem which scales linearly, by using the embeddings to represent discrete probability distributions over the digits. In each iteration, the probability distribution of the carry and the digit classifier are used to calculate the probability distribution of the sum and the next carry. This factorization of the problem is similar to the one used in A-NeSI [32], although we do not approximate and use exact inference. The program and hyperparameters are included in appendix D.1. Results are summarized in table 3.

| Digits per number | 1 | 2 | 4 | 15 | 100 |
|---|---|---|---|---|---|
| LTN [4] | $80.5 \pm 23.3$ | $77.5 \pm 35.6$ | timeout | | |
| NeuPSL [28] | $97.3 \pm 0.3$ | $93.9 \pm 0.4$ | timeout | | |
| DeepProbLog [23] | $97.2 \pm 0.5$ | $95.2 \pm 1.7$ | timeout | | |
| NeurASP [35] | $97.3 \pm 0.3$ | $93.9 \pm 0.7$ | timeout | | |
| DeepStochLog [33] | $97.9 \pm 0.1$ | $96.4 \pm 0.1$ | $92.7 \pm 0.6$ | timeout | |
| Embed2Sym [2] | $97.6 \pm 0.3$ | $93.8 \pm 1.4$ | $91.7 \pm 0.6$ | $60.5 \pm 20.4$ | timeout |
| A-NeSI [32] | $97.7 \pm 0.2$ | $96.0 \pm 0.4$ | $92.6 \pm 0.8$ | $75.9 \pm 2.2$ | overflow |
| DeepSoftLog (Ours) | $\textbf{98.4} \pm 0.1$ | $\textbf{96.6} \pm 0.3$ | $\textbf{93.5} \pm 0.6$ | $\textbf{77.1} \pm 1.6$ | $\textbf{25.6} \pm 3.4$ |
| *Reference* | 98.17 | 96.37 | 92.87 | 75.78 | 15.43 |

Table 3: Test accuracy of the sums on the MNIST addition problem. Results are adapted from [32]. Reference equals $0.9907^{2d}$ where $d$ is the number of digits per number. This is a lower bound on the performance that can be achieved when single-digit supervision is available.

The performance of DeepSoftLog is in line with previous exact systems like DeepStochLog, and we note that the improvement over other exact systems such as DeepProbLog and NeurASP is likely due to their suboptimal hyperparameters. More importantly, DeepSoftLog outperforms the more specialized systems Embed2Sym and A-NeSI, while also scaling further.

| Language | $(01)^*$ | $0^*10^*$ | $(0 \mid 10^*10^*)^*$ |
|---|---|---|---|
| RNN | $77.63 \pm 15.05$ | $61.59 \pm 10.09$ | $50.14 \pm 1.36$ |
| DeepSoftLog | $\mathbf{83.93} \pm 25.87$ | $\mathbf{87.01} \pm 7.18$ | $\mathbf{56.12} \pm 15.98$ |

Table 4: Results for the differentiable automata experiments. We evaluate with the AUC-PR and report the average and standard deviation over 10 seeds.

## 5.3 Differentiable Automata

We can implement a differentiable finite state machine in DeepSoftLog by taking a regular finite state machine in Prolog and making the states and symbols embedded (see listing 1). The resulting model can be trained end-to-end, jointly learning interpretable rules and a perception network.

As an example, consider the regular language $(01)^*$ represented by MNIST images. We learn from positive examples (e.g.  or  ) and negative examples (e.g.  or  ). We train on sequences of lengths up to 4 and test the generalization by evaluating on sequences of length 8 with images not seen during training. Hyperparameters are in appendix D.3.

We pre-train the perception network with 20 images, for which the ground truth image labels are given for each digit. This small amount of concept supervision prevents reasoning shortcuts [25] and eases the optimization problem. As a baseline we use an RNN, which has the same perception network and receives the same pretraining.

Results are summarized in table 4. We repeat the experiment for different regular languages. Not only does DeepSoftLog achieve a higher AUC, but it is also highly interpretable as we can inspect the learned state transitions.

## 6 Related Work

There are several neuro-symbolic systems built on the principle of neural reparametrizations in a symbolic framework. DeepStochLog [33], NeurASP [35], and DeepProbLog [23] are neural extensions of stochastic grammars, answer set programming, and probabilistic logic programming respectively. Lifted Relational Neural Networks [31] and Logic Tensor Networks [4] are based on Datalog with fuzzy semantics (which cannot deal with structured terms) and use the logic program to construct the neural network. These systems make different trade-offs between expressivity and scalability, with fuzzy systems typically being more scalable and probabilistic systems being more expressive.

DeepSoftLog explores an alternative possibility, by neurally reparameterizing the matching mechanism between symbolic symbols instead of the symbols themselves. DeepSoftLog is most closely related to DeepProbLogA* [24], as they both rely on probabilistic possible worlds semantics and backward proving. Hence, it is in principle possible to convert between the two.

The Neural Theorem Prover (NTP) [29] pioneered the association of learnable embeddings with symbols in a logic prover. However, the fuzzy semantics of the NTP leads to gradient sparsity and local minima, as discussed in section 3.1. Furthermore, the NTP lacks several key features of DeepSoftLog like neural functors, recursive rules, and negation. The Greedy Neural Theorem Prover [26] improved the scalability of the NTP. The Conditional Theorem Prover [27] extended the NTP to support rule learning without templates.

The soft matching of symbols in logic is already well-studied in the fuzzy logic programming community [15]. A notable example is Bousi~Prolog [17, 18], which just like the NTP uses soft-unification and fuzzy semantics. These systems still differ considerably from DeepSoftLog as they are not differentiable, and typically rely on the user to manually specify the soft-unification function. WHIRL [6] proposed to add embeddings in DataLog for more flexible data querying. Their soft-join operation can also be seen as an early form of soft-unification, albeit not a learnable one.

Many approaches towards embedding symbols have been investigated for knowledge graphs. Recently, some of these have also been combined with symbolic methods [11, 36]. Generally, these are

less expressive than DeepSoftLog but far more scalable. MINERVA [8] uses reinforcement learning to predict links by walking over a knowledge graph conditional on a query.

## 7 Limitations

DeepSoftLog inherits the advantages of probabilistic logic programming, but also the disadvantages. Probabilistic inference is #P-hard, so exact inference is often not tractable and we need to rely on approximate inference techniques (e.g. [1, 24]). Crucially, some approximate inference methods like $k$-best reintroduce the local minima problem from the NTP. So in future work, we want to consider approximate methods that incorporate a form of exploration, such as sampling methods. What still could be investigated in further work is the use of embeddings to scale up inference in NeSy as in section 5.2.

## 8 Conclusion

We analyzed how a more principled integration of embeddings in logic can be achieved for neuro-symbolic learning, by discussing the properties of learnable soft-unification. We discussed how previous systems do not always satisfy these, and how this can lead to optimization problems. We applied our analysis to create a framework on top of probabilistic logic programming with embeddings, called DeepSoftLog. DeepSoftLog demonstrated that two common methods to neurally extend logic, that of neural facts (as done by DeepProbLog) or neural unification (as done by the NTP) are essentially interchangeable. Lastly, we showed that DeepSoftLog can outperform existing neuro-symbolic methods on both accuracy and scalability.

## Acknowledgments and Disclosure of Funding

This research received funding from the Flemish Government (AI Research Program), the Flanders Research Foundation (FWO) under project G097720N, the KU Leuven Research Fund (C14/18/062) and TAILOR, a project from the EU Horizon 2020 research and innovation program under GA No 952215. Luc De Raedt is also supported by the Wallenberg AI, Autonomous Systems and Software Program (WASP) funded by the Knut and Alice Wallenberg Foundation.

We thank Gabriele Venturato for giving feedback on a draft of this paper. We are also grateful to Giuseppe Marra, Victor Verreet, and Robin Manhaeve for their helpful discussions.

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

# A Proofs

We first state the monotonicity of conjunction, which will be useful later.

**Theorem 6.** *In both fuzzy and probabilistic semantics, it holds that when $c$ is independent of both $a$ and $b$, we have $P(a) \leq P(b) \Rightarrow P(a \wedge c) \leq P(b \wedge c)$.*

*Proof.* In fuzzy semantics, the t-norm is monotonous by definition. Under probabilistic semantics, it follows as $P(a \wedge c) \leq P(b \wedge c)$ reduces to $P(a)P(c) \leq P(b)P(c)$ because of the independencies. $\square$

**Theorem 2.** *Definition 2 holds when the soft-unification function $s$ is $\wedge$-transitive.*

*Proof.* Suppose $b$ is the rest of the proof, such that $P(\pi) = P(b \wedge (t_i \simeq t_j))$ and $P(\pi') = P(b \wedge (t_i \simeq t_k) \wedge (t_k \simeq t_j))$. By the monotonicity of the conjunction and independence between $b$ and the soft-unifications, it now follows that $P(t_i \simeq t_j) \geq P((t_i \simeq t_k) \wedge (t_k \simeq t_j)) \Rightarrow P(\pi) \geq P(\pi')$. Using the soft-unification function the first part becomes $s(t_i, t_j) \geq s(t_i, t_k) \wedge s(t_k, t_j)$, which is exactly the $\wedge$-transitivity rule. $\square$

**Theorem 3.** *Under the fuzzy semiring, definition 3 cannot hold if $s$ is $\wedge$-transitive.*

*Proof.* By the monotonicity of t-norms, $s(x, y) > s(x, z)$ and $s(y, z) > s(x, z)$ implies $s(x, y) \wedge s(y, z) \geq s(x, z) \wedge s(x, z)$. Due to the idempotence of the minimum, this becomes $s(x, y) \wedge s(y, z) \geq s(x, z)$. On the other hand, the $\wedge$-transitivity of these inequalities implies $s(x, y) \wedge s(y, z) < s(x, z)$. This inequality is strict as $s(x, y) = s(y, z) \leq s(x, z)$ implies either $s(x, y) = s(x, z)$ or $s(x, z) = s(x, z)$. $\square$

**Theorem 4.** *If the soft-unification function $s$ is a $\times$-similarity with probabilistic semantics, we satisfy all stated properties (from Def. 1-4).*

*Proof.*

- Def. 1 follows from the well-definedness of the ProbLog semantics, by considering $\simeq$ as a regular probabilistic fact.

- Def. 2 follows from theorem 2 as $s$ is transitive.

- For Def. 3, we can always create a $y$ between some symbols $x$ and $z$, by taking the mean in the embedding space (see theorem 5).

- Def. 4 follows from the symmetry and differentiability of addition and multiplication. Note that we assume that $s$ is differentiable.

$\square$

**Theorem 5.** *A soft-unification function $s$ is a $\times$-similarity, iff $s(x, y) = e^{-d(\mathcal{E}(x), \mathcal{E}(y))}$, where $d$ is a distance function on an embedding space and $\mathcal{E}$ maps symbols to this space.*

*Proof.* We first prove that given that if $s$ is a $\times$-similarity, it can be written as $e^{-d(x,y)}$. We do this by showing that $d(x, y) = -\ln(s(x, y))$ is a distance.

1. As $s(x, x) = 1$, we have that $d(x, x) = -\ln(s(x, x)) = -\ln(1) = 0$.

2. The symmetry of $d$ is implied by the symmetry of $s$:
$$d(x, y) = -\ln(s(x, y)) = -\ln(s(y, x)) = d(y, x)$$

3. The triangle inequality can be proven from the $\times$-transitivity, and the monotonicity of $\ln$:
$$d(x, z) = -\ln(s(x, z)) \leq -\ln(s(x, y) \cdot (y, z)) = d(x, y) + d(y, z)$$

As we do not have distinguishability (there can be a $x \neq y$ such that $d(x, y) = 0$), the embedding space is a pseudometric. The other direction can be proven in a similar fashion, by evaluating that $e^{-d(x,y)}$ satisfies reflexivity, symmetry, and $\times$-transitivity. $\square$

## B   Example programs

We implement a variation on the alarm basian network in DeepSoftLog where the earthquake and landslide symbols are embedded. On the right, we show the translated ProbLog program.

$0.2 :: \mathtt{event}(\sim\mathtt{earthquake}).$

$0.5 :: \mathtt{hears\_alarm}(\mathtt{mary}).$

$\mathtt{alarm} :- \mathtt{event}(\sim\mathtt{landslide}).$

$\mathtt{calls}(\mathtt{X}) :- \mathtt{alarm}, \mathtt{hears\_alarm}(\mathtt{X}).$

$0.2 :: \mathtt{event}(\sim\mathtt{V}_1) :- \sim\mathtt{V}_1 \simeq \sim\mathtt{earthquake}.$

$0.5 :: \mathtt{hears\_alarm}(\mathtt{mary}).$

$\mathtt{alarm} :- \mathtt{event}(\sim\mathtt{landslide}).$

$\mathtt{calls}(\mathtt{X}) :- \mathtt{alarm}, \mathtt{hears\_alarm}(\mathtt{X}).$

$0.5 :: \sim\mathtt{landslide} \simeq \sim\mathtt{earthquake}.$

$0.5 :: \sim\mathtt{earthquake} \simeq \sim\mathtt{landslide}.$

$\sim\mathtt{landslide} \simeq \sim\mathtt{landslide}.$

$\sim\mathtt{earthquake} \simeq \sim\mathtt{earthquake}.$

Next, we also demonstrate an example with a linear rule.

$\mathtt{eq}(\mathtt{X}, \mathtt{X}).$

$\mathtt{q} :- \mathtt{eq}(\mathtt{f}(\sim\mathtt{a})), \mathtt{f}(\sim\mathtt{b})).$

$\mathtt{eq}(\mathtt{V}_1, \mathtt{V}_2) :- \mathtt{V}_1 \simeq \mathtt{V}_2.$

$\mathtt{q} :- \mathtt{eq}(\mathtt{f}(\sim\mathtt{a})), \mathtt{f}(\sim\mathtt{b})).$

$\mathtt{f}(\mathtt{V}_1) \simeq \mathtt{f}(\mathtt{V}_1) :- \mathtt{V}_1 \simeq \mathtt{V}_2.$

$\mathtt{a} \simeq \mathtt{a}.$

$\mathtt{b} \simeq \mathtt{b}.$

$\sim\mathtt{a} \simeq \sim\mathtt{a}.$

$0.1 :: \sim\mathtt{a} \simeq \sim\mathtt{b}.$

$0.1 :: \sim\mathtt{a} \simeq \sim\mathtt{f}(\mathtt{a}).$

$0.1 :: \sim\mathtt{a} \simeq \sim\mathtt{f}(\mathtt{b}).$

$\sim\mathtt{b} \simeq \sim\mathtt{b}.$

$0.1 :: \sim\mathtt{b} \simeq \sim\mathtt{f}(\mathtt{b}).$

$0.1 :: \sim\mathtt{b} \simeq \sim\mathtt{f}(\mathtt{a}).$

$\sim\mathtt{f}(\mathtt{a}) \simeq \sim\mathtt{f}(\mathtt{a}).$

$0.1 :: \sim\mathtt{f}(\mathtt{a}) \simeq \sim\mathtt{a}.$

$0.1 :: \sim\mathtt{f}(\mathtt{a}) \simeq \sim\mathtt{b}.$

$0.1 :: \sim\mathtt{f}(\mathtt{a}) \simeq \sim\mathtt{f}(\mathtt{b}).$

$\sim\mathtt{f}(\mathtt{b}) \simeq \sim\mathtt{f}(\mathtt{b}).$

$0.1 :: \sim\mathtt{f}(\mathtt{b}) \simeq \sim\mathtt{a}.$

$0.1 :: \sim\mathtt{f}(\mathtt{b}) \simeq \sim\mathtt{b}.$

$0.1 :: \sim\mathtt{f}(\mathtt{b}) \simeq \sim\mathtt{f}(\mathtt{a}).$

## C   Semantics

To prove the equivalence of the transformation with soft-unification, we need to prove two separate facts. First that if two terms soft-unify, they will unify in the transformed program. Second that they will produce the same soft-unification facts.

**Theorem 6.** *Consider two DeepSoftLog expressions $a$ and $b$, and the ProbLog expression $a^*$ which is the translation of $a$. Now $a$ will soft-unify with $b$ if and only if $a^*$ unifies with $b$ (using regular unification). Furthermore, they will result in the same substitution (ignoring the extra introduced variables), and the same soft-unifications.*

*Proof.* We prove the theorem by structural induction on $a$.

    1. If $a$ is a variable or constant, we have $a = a^*$, so it holds trivially.

2. If $a$ is an embedded term, we have that $a^* = \sim V$, where $V$ is a fresh variable. Hence both $a$ will unify with $b$ iff $b$ is embedded. If $b$ is embedded, both will add the soft-unification fact $b \simeq a$, as the translated rule will have added this in the body.

3. If $a$ is a functor, we have that $a = f(t_1, ..., t_k)$ and $a^* = f(t_1^*, ..., t_k^*)$. If $b$ is not of the form $f(s_1, ..., s_k)$, both the soft-unification and unification will fail. In the other case, they will both give the same result as the unifications between $s_1$ and $t_1$ or $t_1^*$ give the same result by structural induction. As rules are linear, variables of the $t_i$ and $t_j$ do not overlap, meaning that the substitutions can be combined without conflicts.

4. If $a$ is an atom (i.e. a rule head), we can use an identical proof as for functors.

Suppose $t_1$ and $t_2$ is the corresponding term in the translated program. If $t_1$ is a variable or a non-embedded term, we have $t_1 = t_2$, which means both cases will result in the same substitution when unified with a third term $t_3$. In the case that $t_1$ is an embedded term, $t_2$ will be of the form $\sim V$ where $V$ is a variable. Hence, when $t_3$ is not embedded, the soft-unification with $t_1$ will fail, and the regular unification with $t_2$ will also fail. When $\sim t_3$ is embedded, the soft-unification with $s_2$ succeeds and regular unification also succeeds with the substitution $V \to t_3$. The substitutions returned by soft-unification are not exactly the same as in regular unification, because of the introduced variables. However, as these are unique and do not appear in the DeepSoftLog program, they can be safely ignored. □

The above analysis can also be extended to the case of non-linear rules (see the section on linearization).

# D   Experiment details

## D.1   MNIST-addition

The code for MNIST-addition is given in listing 2. Hyperparameters are summarized in table 5. Of note in this example is that we encode as much background knowledge as possible. The green cut encodes the independency of the sum of lower digits from the higher digits, so we can find the 1-best proof faster during evaluation (an alternative solution would be to use a geometric mean heuristic [24]). The embedded functors `mod_ten_add` and `carry` could be learned by neural networks. However, for maximal performance, we hand-coded them. This is possible as we know what the probability distribution of the modulo addition and carry should be.

Listing 2: Code for the MNIST addition experiments

```
digit(X) :- member(X, [0, 1, 2, 3, 4, 5, 6, 7, 8, 9]).
embed_digit(EMB, DIGIT) :- digit(DIGIT), eq(~DIGIT, EMB).
eq(X, X).

add(X,Y,Z) :- add_(X, Y, Z, ~0).

add_([], [], [], ~0).
add_([], [], [1], ~1).
add_([HX|TX], [HY|TY], [HZ|TZ], CARRY) :-
    embed_digit(~mod_ten_add(HX, HY, CARRY), HZ),
    !, % green cut for faster evaluation
    add_(TX, TY, TZ, ~carry(HX, HY, CARRY)).
```

We ran the experiments on CPU (Intel® Core™ i7-2600 CPU @ 3.40GHz) with 16GB of RAM.

## D.2   Countries

Hyperparameters are summarized in table 6. We ran all experiments on a single CPU (Apple M2).

| | |
|---|---|
| optimizer | AdamW |
| learning rate | 0.0003 |
| learning rate schedule | cosine |
| training epochs | 100 |
| weight decay | 0.00001 |
| batch size | 4 |
| embedding dimensions | 10 |
| embedding initialization | one-hot, fixed |
| neural networks | LeNet5 |
| max search depth | / |

Table 5: Hyperparameters for the MNIST-addition experiments.

| | |
|---|---|
| optimizer | AdamW |
| learning rate | 0.01 |
| learning rate schedule | cosine |
| training epochs | 6 |
| weight decay | 0 |
| batch size | 4 |
| embedding dimensions | 100 |
| embedding initialization | uniform over hypersphere |
| neural networks | / |
| max search depth | 2 (S1, S2) or 3 (S3) |
| max branching factor | 4 |

Table 6: Hyperparameters for the countries experiments.

## D.3 Differentiable Turing Machine

Hyperparameters are summarized in table 3. We ran all experiments on a single CPU (Apple M2).

| | |
|---|---|
| optimizer | AdamW |
| learning rate embeddings | 0.1 |
| learning rate perception | 0.0001 |
| learning rate schedule | cosine |
| training epochs | 25 |
| weight decay | 0.00001 |
| batch size | 8 |
| embedding dimensions | 3 |
| embedding initialization | uniform over hypersphere |
| neural networks | LeNet5 |
| max search depth | / |

Table 7: Hyperparameters for the differentiable finite state machine experiment.

## E Embeddings as fuzzy logic

A different perspective on embeddings from what we considered in this paper, is to see them as a discrete distribution (i.e. the embedding is a probability vector). When we make sure embeddings are normalized (i.e. positive and sum to one) and take the dot product as a soft-unification function, we essentially get the probability that two embeddings are equal. By conjoining different soft-unifications, we have a fuzzy interpretation, as the soft-unifications are assumed to be independent.

As a demonstration, we apply this to the visual sudoku problem [3]. This benchmark requires classifying if a grid of images constitutes a valid sudoku puzzle. We follow the same protocol as in [32]. Hyperparameters for the visual sudoku experiment are in table 9. The learning rate, weight decay, and gradient clipping were chosen by Bayesian optimization on the 11th validation split. We averaged the results over the other 10 splits. We ran all experiments on CPU (Intel(R) Xeon(R) CPU

| Visual Sudoku | $4 \times 4$ | $9 \times 9$ |
| --- | --- | --- |
| CNN | $51.5 \pm 3.34$ | $51.2 \pm 2.20$ |
| NeuPSL [28] | $89.7 \pm 2.20$ | $51.5 \pm 1.37$ |
| A-NeSI [32] | $89.8 \pm 2.08$ | $62.3 \pm 2.20$ |
| DeepSoftLog | $\mathbf{94.2} \pm 1.84$ | $\mathbf{65.0} \pm 1.94$ |

Table 8: Accuracy on visual sudoku classification. Previous results are adapted from [32].

E3-1225 v3 @ 3.20GHz). The $4 \times 4$ and $9 \times 9$ runs took about 95 and 630 seconds per experiment respectively.

Table 8 summarizes the results of DeepSoftLog and compares them with current state-of-the-art. Surprisingly, the crude fuzzy approximation outperforms all existing systems by a considerable margin.

| | $4 \times 4$ | $9 \times 9$ |
| --- | --- | --- |
| optimizer | AdamW | AdamW |
| learning rate | 0.00162 | 0.000671 |
| training epochs | 100 | 300 |
| weight decay | 0.0000144 | 0.000116 |
| batch size | 1 | 1 |
| gradient clipping | 2.445 | 2.753 |
| embedding dimensions | 10 | 10 |

Table 9: Hyperparameters for the visual sudoku experiments.

