# OpenReview forum: "Soft-Unification in Deep Probabilistic Logic"
_NeurIPS.cc/2023/Conference — NeurIPS 2023 poster_

### Official Review · Reviewer_bQCz · 2023-07-05

**Soundness:** 3 good
**Presentation:** 2 fair
**Contribution:** 3 good
**Rating:** 6
**Confidence:** 4

**Summary:**

This work proposes a neural symbolic framework, DeepSoftLog, which extends DeepProbLog with a soft equivalent operation.
The authors also develops four properties that such a soft equivalency operation should hold.
The experimental results demonstrate that DeepSoftLog has better performance than the state-of-the-art baselines.

**Strengths:**

Originality: 2/5
The idea of soft equivalency is not new, which has been adopted by works such as NTP.
Although the authors raise a novel suite of 4 standards that a good soft equivalency function should enjoy, they have not developed deep enough from these definitions, to discuss the theoretical benefits that these properties could bring, such as convergence, relaxation, and learning efficiency.

Quality: 2/5
Pros: The authors provide proof that the soft equivalency function satisfies all four properties.
Cons: In the experiment section, all images are encoded with one-hot embedding, which is quite problematic. This embedding not only creates an unfair comparison against the baselines but is also not meaningful for future references.

Clarity: 3/5
Pros: The writing is pretty clear and the four properties are quite easy to understand.
Cons: The program syntax is hard to understand and not quite user-friendly. For example, p(~x):- q(~y) is not intuitive to understand, as y is not used in the head, and a new, unknown variable is introduced in the context.

Significance: 1/5
The theory part is not developed in-depth enough, which undermines the necessity of why the properties are required.
Further, although the experiment results outperform the SOTA quite a lot, the embedding is synthesized, which also undermines the credibility of the result.

**Weaknesses:**

See "Strength" section.

**Questions:**

Can you address what are the benefits that the four properties bring?

**Limitations:**

Yes.

---

> ### Author Rebuttal · Authors · 2023-08-09
>
> We sincerely thank the reviewer for taking the time to read and review our paper. We will address the reviewer's comments, ordered by topic.
>
> > In the experiment section, all images are encoded with one-hot embedding, which is quite problematic.
>
> The authors want to stress that the images are embedded by a neural network and are _not_ one-hot encoded. Perhaps the statement in appendix D1 was confusing, which says that the digits for the (symbolic) ground truth labels are represented as one-hot vectors. We will clarify this sentence.
>
> > Further, although the experiment results outperform the SOTA quite a lot, the embedding is synthesized, which also undermines the credibility of the result.
>
> We stress again that the embeddings in all experiments are learned from data by gradient descent. This is stated e.g. in the introduction when we say that we are "using learnable embeddings".
>
> > "The program syntax is hard to understand and not quite user-friendly."
>
> We adopt the syntax of the Prolog language (with some very minor additions), which we summarize in section 2.1. Prolog is by far the most popular and well-known language to represent logic knowledge and has been extensively used for this purpose. We therefore feel it is a natural choice for this paper, as it builds on the probabilistic and neural extensions of Prolog: ProbLog and DeepProbLog.
>
> > "For example, p(x):- q(y) is not intuitive to understand, as y is not used in the head, and a new, unknown variable is introduced in the context."
>
> We follow the Prolog convention where lowercase symbols are constants (see section 2.1). So `x` and `y` are not variables here. The rule `p(x) :- q(y).` means that if the atom `q(y)` is true, the atom `p(x)` is also true.
>
> > "The theory part is not developed in-depth enough, which undermines the necessity of why the properties are required."
> > "Although the authors raise a novel suite of 4 standards that a good soft equivalency function should enjoy, they have not developed deep enough from these definitions, to discuss the theoretical benefits that these properties could bring, such as convergence, relaxation, and learning efficiency."
>
> We address the specific points that are mentioned.
> - (Convergence) Giving proper convergence proofs for gradient-based optimization is very challenging, and often even impossible for neural network based systems [1]. We note that the related frameworks in neuro-symbolic AI also do not provide this. We do discuss local minima and gradient flow of our method and compare it with previous fuzzy methods (see e.g. lines 151-158 and 199-203).
> - (Relaxation) Soft-unification is indeed a relaxation of the regular (i.e. hard) logic. This is known and has been shown in previous work [2].
> - (Learning efficiency) We do make claims about the learning efficiency. Most crucially, our method gives gradients to all embeddings during a training step (disregarding critical points), while previous methods which used the Godel t-norm only give a gradient to at most 2 embeddings. We also demonstrate this empirically, see the ablations in figure 2.
> If there are other concrete points where the reviewer feels our theoretic analysis is lacking, we would be happy to address those.
>
> > Can you address what are the benefits that the four properties bring?
>
> We summarize the important benefits. An in-depth discussion can be found in section 3.
> - DeepSoftLog optimizes better compared to previous systems (e.g. NTP). This is mostly due to satisfying Def4. It has previously been shown [1] that gradient descent gets stuck in local minima if not all proof paths get updated simultaneously (see lines 151-158).
> - Def2 also improves the training stability by disallowing redundancy in the proof paths (see also the ablations in figure 2).
> - Soft-unification has so far been only used in fuzzy systems, while DeepSoftLog equips probabilistic semantics. The use of probabilities as opposed to fuzzy values gives performance advantages. Note that this is the only aspect that impacts the computational complexity (see section 7). In summary, the computational complexity is the same as for ProbLog. The trade-off of fuzzy and probabilistic logic between performance and scalability is well-known in the literature.
> - Def3 has previously been found to be necessary to make the soft-unification sufficiently expressive [4].
>
> [1]: Swirszcz, Grzegorz, Wojciech Marian Czarnecki, and Razvan Pascanu. "Local minima in training of neural networks." arXiv preprint arXiv:1611.06310 (2016).
> [2]: Sessa, Maria I. "Approximate reasoning by similarity-based SLD resolution." Theoretical computer science 275.1-2 (2002): 389-426.
> [3]: de Jong, Michiel, and Fei Sha. "Neural theorem provers do not learn rules without exploration." arXiv preprint arXiv:1906.06805 (2019).
> [4]: Julián-Iranzo, Pascual, Clemente Rubio-Manzano, and Juan Gallardo-Casero. "Bousi~Prolog: a Prolog extension language for flexible query answering." Electronic Notes in Theoretical Computer Science 248 (2009): 131-147.

---

> > ### Comment · Reviewer_bQCz · 2023-08-21
> >
> > I have double check the implementation in the supplementary material, and the embeddings are pictures instead of the one-hot embeddings as claimed in the Appendix D1. I will raise my rating accordingly.

---

### Official Review · Reviewer_4neU · 2023-07-06

**Soundness:** 3 good
**Presentation:** 4 excellent
**Contribution:** 3 good
**Rating:** 7
**Confidence:** 4

**Summary:**

This paper studies the notion of *soft-unification*, first employed by the Neural Theorem Prover to learn logic rules in an end-to-end differentiable manner. They outline several properties that need to hold for soft-unification to be semantically meaningful and efficiently trainable; properties which previous frameworks fail to satisfy. Consequently, the authors introduce a framework satisfying such properties, which they term *DeepSoftLog*, which is essentially an integration of embeddings (as opposed to symbols) with probabilistic logic programming.

**Strengths:**

- While the idea of soft-unification is certainly not new, the authors tackle the problem in a principled manner by integrating
embeddings with problog, a language for probabilistic logic programming.

- The paper is very well written for the most part, with the authors using examples throughout to help elucidate the exposition.

- While doing a good job with exposition, the authors also managed to stay rigorous throughout the paper, with definition and theorems throughout.



**Weaknesses:**


- I find the paper to be lacking in terms of experimental evaluation. The experimental section does not offer much that convinces me of the merits of proposed approach:

  - The advantage of using probabilistic semantics as opposed to fuzzy semantics is well documented in the literature, which seems to be the main conclusion of section 5.1.

  - I'm not really sure what the point of evaluating on MNIST-addition is. The authors seems to be using it to argue for the scalability of their approach. However, I don't see how their approach is more scalable compared to DeepProbLog since they incur an extra branching factor owing soft-unification. They mention the use of approximate inference techniques, but then they have to deal with the pitfalls of sparse gradients, but is understandable given the intractability of exact inference. Furthermore, since such approximations were not proposed by the authors, it seems only fair to also utilize them when comparing against DeepProbLog.

  - Section 5.3 is to me, maybe what this entire paper is all about. However, I feel it would need to be expanded considerably.

- There has been a lot of work on Neuro-Symbolic AI, and in my opinion, the related works section does not do it justice.

Typos:
- I believe line 87 should read as "(2) Disjoin the proofs..."

**Questions:**

- Could you please explain what you meant by the paragraph on lines 204-208?
- On lines 219-221 you mention that you need a neural network for every functor. I'm guessing this does not scale if you have a large logic program?
- On lines 247-248 you mention that you "do not actually ground the soft-unification function s, but provide it as a built-in". Could please clarify what you mean by that? It is my understanding that when compiling into a circuit, everything is grounded?
- On lines 259-260 you mention that "An important constraint we apply is the soft-unification predicate needs to be ground upon evaluation". Could you clarify what that means? Is that not at odds with the statement in the above question?

**Limitations:**

I believe the authors have adequately stated the limitations.

---

> ### Author Rebuttal · Authors · 2023-08-09
>
> We want to sincerely thank the reviewer for taking the time to read and review our paper. We are happy to hear that the reviewer found the paper very well written.
>
> > I'm not really sure what the point of evaluating on MNIST-addition is. The authors seems to be using it to argue for the scalability of their approach. However, I don't see how their approach is more scalable compared to DeepProbLog since they incur an extra branching factor owing soft-unification. They mention the use of approximate inference techniques, (...). Furthermore, since such approximations were not proposed by the authors, it seems only fair to also utilize them when comparing against DeepProbLog.
>
> We will try to make this more clear in the paper.
> - First, we stress that the MNIST-addition experiment does not use approximate inference techniques but is exact (see line 299). Our solution to MNIST-addition uses a different encoding compared to DeepProbLog, which avoids the combinatorial explosion for higher digits.
> - The new, more efficient encoding relies on embeddings. When such an encoding can be used, this "tensorization" is asymptotically faster (comparable to lifted inference vs ground inference). You are right that this could also be done in DeepProbLog if the implementation supports embeddings and neural functors, but this would just give you a very similar system to DeepSoftLog.
> - The authors believe this experiment is relevant to the community as it shows that MNIST-addition is solvable exactly in linear time in the number of digits, which has not been demonstrated before. This suggests that the use of embeddings in neuro-symbolic methods can be very powerful.  It also suggests that the MNIST-addition experiment (one of the most common neuro-symbolic benchmark) actually rather easy when embeddings can be used.
>
> > Section 5.3 is to me, maybe what this entire paper is all about. However, I feel it would need to be expanded considerably.
>
> We agree that this experiment is the most interesting setting for DeepSoftLog. We have since performed experiments on more grammars and added a neural baseline (You can find the expanded section 5.3 in the pdf attached to the Author Response). On the other hand, these experiments are limited significantly due to the scalability of knowledge compilation and would require approximation techniques to properly scale up, which we leave for further research.
>
> > Could you please explain what you meant by the paragraph on lines 204-208?
>
> In this paragraph, we motivate our concrete choice for the soft-unification function s. Theorem 5 gives us the exponential form for s, but the choice of distance function d is still open. We use the angular distance (i.e. arcos of cosine similarity) as this optimizes well. But as the soft-unification function is the negative exponential of the distance function (theorem 5), this would mean that the minimal soft-unification is achieved when two vectors lie opposite on the sphere. In other words, we can pack very few symbols in our embedding space, before they all start to soft-unify with each other, which is undesirable (creates big branching factors, or requires high embedding dimensions). By taking an absolute value, the minimum soft-unification is achieved by orthogonal vectors, which means we get exponentially more positions with low soft-unification.
>
> > On lines 219-221 you mention that you need a neural network for every functor. I'm guessing this does not scale if you have a large logic program?
>
> If you have a very large number of different functors, this could indeed become a problem. In our experience, this has not been an issue because in practice the more pressing concern for scalability is the probabilistic inference.
>
> > On lines 247-248 you mention that you "do not actually ground the soft-unification function s, but provide it as a built-in". Could please clarify what you mean by that? It is my understanding that when compiling into a circuit, everything is grounded?
>
> You are correct that when compiling a circuit, everything is grounded. What we mean is that we do not explicitly create the full grounding of the soft-unification function, as most of these soft-unifications would not be used. Instead, our implementation provides a built-in that generates these facts as needed.
>
> Compare this with how in ProbLog, a first-order rule also has a large (or infinite) number of groundings, but only the subset that is actually needed for the query is generated  (a concept known as the relevant grounding).
>
> > On lines 259-260 you mention that "An important constraint we apply is the soft-unification predicate needs to be ground upon evaluation". Could you clarify what that means?
>
> Consider this DeepSoftLog program:
>
> ```
> p(~a).
> query :- p(~X).
> ```
>
> In this example, if we query `query`, the variable `X` does not get instantiated before we arrive at the soft-unification (between `~a` and `~X`). This is still well-defined: `X` could be instantiated with every possible element in the domain, but this usually results in an infinite amount of unifications. To avoid this problem, we constrain our inference to cases where this X would be already instantiated. Otherwise, we throw a runtime error.
>
> Note that this is very similar to regular ProbLog, where you can have first-order probabilistic facts, as long as the grounding with respect to the query is finite.
>
> > I believe line 87 should read as "(2) Disjoin the proofs..."
>
> We will correct this mistake, thank you for noticing.

---

> > ### Comment · Reviewer_4neU · 2023-08-14
> >
> > Thank you for you response. I believe your response provides a satisfactory answer to all of my questions. I will raise my score.

---

### Official Review · Reviewer_nJ1y · 2023-07-17

**Soundness:** 3 good
**Presentation:** 3 good
**Contribution:** 3 good
**Rating:** 6
**Confidence:** 3

**Summary:**

This work proposes DeepSoftLog, a neuro-symbolic framework that generalizes ProbLog by combining soft-unification with probabilistic semantics. It first defines some properties of the soft-unification that it believes to be required for meaningful semantics and efficient training. Further, it examines some of the existing neuro-symbolic systems and shows that they satisfy a few but not all of the defined properties. In contrast, the proposed DeepSoftLog is able to satisfy all the properties. In the empirical evaluations, three experiments are performed where the proposed DeepSoftLog is able to outperform the baselines for the first two.

**Strengths:**

The theoretical analysis of learnable soft-unification is technically sound and is potentially interesting to the other neuro-symbolic systems where unification is present. From the first two experiments, the proposed DeepSoftLog achieves significant improvements and also ablation study on the proposed soft-unification properties is included. The paper has provided sufficient background for the readers to understand this work.

**Weaknesses:**

- My main concern is that the technical details are not clear to me. From Algorithm 1, it is unclear how the score in soft unification is set; how the soft-unification is incorporated during inference and knowledge compilation step is not explained while this is key to understanding the probabilistic semantics of soft-unification.
- Another concern is that the motivation for the soft-unification is not adequately justified. I want to see more about why the defined four properties can improve performance and why soft unification with the defined four properties can outperform the other neuro-symbolic systems. Also, I wonder how these properties affect computational complexity.
- At Line 165, both s(x, y) and s(y, z) are real numbers in [0, 1]. I wonder what a conjunction of two numbers means.
- Both \land-transitive and \times-similarity are not defined, while they are used in Theorem 2&4 respectively.
- In Definition 3, should it be that x \neq z.
- Authors should justify why there's no baseline included in the differentiable finite state machine experiment in Sec. 5.3 to make the empirical evaluation thorough.

**Questions:**

- Can the authors provide more motivations on why the four properties of soft-unification can improve performance and some analysis on how they affect complexity?
- Can the authors explain how the soft-unification scores are incorporated during inference?
- I don't understand the example in the introduction: what is the meaning of Listing 1, what are the ~newstate1 & ~newstate4, and why DeepSoftLog would learn to set ~newstate1 = ~state2 & ~newstate4 = ~state1?
- Why there's no baseline for experiment in Sec. 5.3?

**Limitations:**

Yes.

---

> ### Author Rebuttal · Authors · 2023-08-09
>
> We first sincerely thank the reviewer for taking the time to read and review our paper.
>
> > I don't understand the example in the introduction: what is the meaning of Listing 1, what are the ~newstate1 & ~newstate4, and why DeepSoftLog would learn to set ~newstate1 = ~state2 & ~newstate4 = ~state1?
>
> First we note that ~new_state1 and ~new_state4 on line 39 should have been ~prev_state1 and ~prev_state2 respectively. We apologize for this typo.
>
> The example in Listing 1 implements a finite state machine in DeepSoftLog. However, the states (e.g. ~state1 or ~prev_state2) are not purely symbolic but are represented with embeddings. Hence DeepSoftLog can learn the transitions of the automata from data. In this example, DeepSoftLog can create a transition from state 1 to state 2 by assigning the same embedding to ~prev_state1 and ~state2. During training, we optimize these embeddings by giving positive and negative examples, and minimizing the cross-entropy loss with gradient descent (see experiment in section 5.3, where the automata is learned jointly with the perception).
>
> > My main concern is that the technical details are not clear to me. From Algorithm 1, it is unclear how the score in soft unification is set; how the soft-unification is incorporated during inference
> > Can the authors explain how the soft-unification scores are incorporated during inference?
>
> A main contribution of the paper is that soft-unification can be transformed into probabilistic logic by explicitly encoding the soft-unifications as probabilistic facts. As an example, suppose we have the following DeepSoftLog program:
>
> ```
> p(~a)
> query :- p(~b).
> ```
>
> Then we can transform it into an equivalent ProbLog program:
>
> ```
> 0.5 :: ~a ≃ ~b.
> p(X) :- ~a ≃ X.
> query :- p(~b)
> ```
>
> So after this transformation, which introduces the probabilistic fact  ~a ≃ ~b to implement soft unification,  we can do regular probabilistic inference. We explain this transformation more formally on lines 235-267. We also prove that this is equivalent to regular soft-unification (proof is in appendix C). Some more elaborate examples of this transformation are included in appendix B.
>
> > knowledge compilation step is not explained while this is key to understanding the probabilistic semantics of soft-unification.
>
> We first want to stress that knowledge compilation (KC) is unrelated to the probabilistic semantics, but is the standard way to do probabilistic inference in probabilistic logics such as Problog. We chose not to introduce KC because (1) understanding it is not really relevant to the content of the paper, besides solving the disjoint sum problem, and (2) KC would require a fairly lengthy introduction. We refer to [3] for more details.
>
> > At Line 165, both s(x, y) and s(y, z) are real numbers in [0, 1]. I wonder what a conjunction of two numbers means.
>
> How the conjunction is evaluated depends on the semantics. So e.g. for probabilistic logic, this conjunction is evaluated as a multiplication. This is explained in lines 166-167.
>
> > Both $\land$-transitive and $\times$-similarity are not defined, while they are used in Theorem 2&4 respectively.
>
> The definitions of $\land$-transitivity and $\times$-similarity can be found on lines 165 and 168 respectively.
>
> > Another concern is that the motivation for the soft-unification is not adequately justified. I want to see more about why the defined four properties can improve performance and why soft unification with the defined four properties can outperform the other neuro-symbolic systems.
> > Can the authors provide more motivations on why the four properties of soft-unification can improve performance and some analysis on how they affect complexity?
>
> We summarize the main points of how DeepSoftLog concretely improves the performance and how this relates to the proposed properties:
> - DeepSoftLog optimizes better compared to previous systems (e.g. NTP). This is mostly due to satisfying Def4. It has previously been shown [1] that gradient descent gets stuck in local minima if not all proof paths are updated simultaneously (see lines 151-158).
> - Def2 also improves the training stability by disallowing redundancy in the proof paths (see also the ablations in figure 2).
> - Soft-unification has so far been only used in fuzzy systems, while DeepSoftLog is based on probabilistic semantics. The use of probabilities as opposed to fuzzy values gives DeepSoftLog a performance advantage. Note that this is the only aspect that impacts the computational complexity (which is the same as for ProbLog, see section 7). This trade-off of fuzzy and probabilistic logic between performance and scalability is well-known in the literature.
> - Def3 has previously been found to be necessary to make the soft-unification sufficiently expressive [2].
> For a more in depth discussion, we refer to section 3.
>
> > Why there's no baseline for experiment in Sec. 5.3?
>
> This is a good point, and we also considered it. However, we are unaware of an existing neuro-symbolic framework that could properly implement this experiment, as it requires both learnable perception and learned structure. We have since implemented an RNN as a simple neural baseline. You can find the results in the pdf attached to the “Author Rebuttal”.
>
> > In Definition 3, should it be that x \neq z.
>
> We will correct this typo, thank you for noticing.
>
> [1]: de Jong, Michiel, and Fei Sha. "Neural theorem provers do not learn rules without exploration." arXiv preprint arXiv:1906.06805 (2019).
> [2]: Julián-Iranzo, Pascual, Clemente Rubio-Manzano, and Juan Gallardo-Casero. "Bousi~ Prolog: a Prolog extension language for flexible query answering." Electronic Notes in Theoretical Computer Science 248 (2009): 131-147.
> [3]: Fierens, Daan, et al. "Inference and learning in probabilistic logic programs using weighted boolean formulas." Theory and Practice of Logic Programming 15.3 (2015): 358-401.

---

### Author Rebuttal · Authors · 2023-08-09

As was requested by some reviewers, we have expanded section 5.3 with additional experiments and a neural baseline. We have attached the new section 5.3 as a pdf.

---

### Decision · Program_Chairs · 2023-09-21

**Decision:**

Accept (poster)

**Comment:**

First of all, thanks for understanding we are one review short on this one. It is an exceptional circumstance. After careful discussion, we are clear on this paper’s acceptance. Though soft unification of logic is certainly not a novel idea in this community, the reviews and senior reviewers all agree that this paper is rigorous and the proposed method is insightful. Still, we would like to note two aspects for further improvements. First of all, the empirical results are not convincing enough. Though MNIST-addition is a well-adopted setting in our community, the benefit of adopting this experiment is not really in favor of this paper. The reviews suggest that this argues for scalability of the proposed method. However, the empirical results do not fully back up this claim. We would highly recommend this paper to continue exploration an additional experiment setting. It is not just this paper’s duty, but we as the whole community needs to keep working on it so that principled neuro-symbolic methods can be more well accepted into other domains. Secondly, the theoretical parts can be more elaborate. The current analysis does not justify the necessaries of the four properties. What exactly benefits do those four properties bring to this method, and where can this benefit be fully demonstrated?